# Vitamin B12 in Cancer Patients: Clinical Insights into Deficiency, Excess, Diagnosis, and Management

**DOI:** 10.3390/nu17203272

**Published:** 2025-10-17

**Authors:** Małgorzata Osmola, Martyna Tyszka, Adam Jirka, Olga Ciepiela, Aleksandra Kapała, Marco Vincenzo Lenti, Tamara Matysiak-Budnik

**Affiliations:** 1Maria Sklodowska-Curie Medical Academy in Warsaw, 03-411 Warsaw, Poland; 2Department of Clinical Oncology, Medical University of Warsaw, 02-091 Warsaw, Poland; 3IMAD, Hepato-Gastroenterology & Digestive Oncology, University Hospital of Nantes, 44-000 Nantes, France; 4Department of Laboratory Medicine, Medical University of Warsaw, 02-091 Warsaw, Poland; 5Department of Oncology Diagnostics, Cardio-Oncology and Palliative Medicine, Maria Skłodowska-Curie National Research Institute of Oncology, 02-871 Warsaw, Poland; aleksandra.kapala@nio.gov.pl; 6Department of Internal Medicine and Medical Therapeutics, University of Pavia, 27-100 Pavia, Italy; 7First Department of Internal Medicine, Fondazione IRCCS Policlinico San Matteo, 27-100 Pavia, Italy; 8CHU de Nantes, INSERM, Center for Research in Transplantation and Translational Immunology, UMR 1064, Institut de Transplantation Urologie-Néphrologie, Nantes Université, 44-000 Nantes, France

**Keywords:** vitamin B12 deficiency in cancer, hypercobalaminemia, vitamin B12 supplementation in cancer, homocysteinemia in cancer, methymalonic acid, vitamin B12 in CIPN

## Abstract

Vitamin B12 (cobalamin) is a critical micronutrient involved in hematopoiesis and neurological function. Its deficiency, commonly presenting with anemia and neurological symptoms, is particularly relevant in oncology. While anemia affects up to 60% of cancer patients, the contribution of vitamin B12 deficiency to cancer-related anemia remains underexplored. Additionally, cobalamin-related neuropathy manifests or exacerbates existing chemotherapy-induced peripheral neuropathy (CIPN), a serious side effect of chemotherapy. Prevalence estimates in cancer populations range widely (6–48%), with higher rates in elderly and gastrointestinal cancer patients. This review summarizes current evidence on the prevalence and implications of both vitamin B12 deficiency and excess in patients with solid tumors. It discusses laboratory markers (such as serum vitamin B12, holotranscobalamin, methylmalonic acid, and homocysteine) that could improve diagnostic accuracy in oncology settings. Additionally, it evaluates supplementation strategies and discusses its role in mitigating CIPN. Additionally, it addresses B12′s emerging immunological role in cancer therapy.

## 1. Introduction

Vitamin B12, also known as cobalamin, is a micronutrient essential for red blood cell production and neurological function. At the cellular level, vitamin B12 is required as a coenzyme for the methyl transfer reaction for two metabolic enzymes: methionine synthase and L-methylmalonyl-coenzyme A mutase [1,2].

The most common symptoms of vitamin B12 deficiency are anemia and neurological manifestations, including sensory peripheral neuropathy. These symptoms are important in oncological patients, since anemia is reported in around 30–60% of cancer patients [3,4]. Still, there is limited data about vitamin B12 deficiency contributing to cancer-related anemia. On the other hand, sensory peripheral neuropathy caused by vitamin B12 deficiency is similar to or can aggravate existing chemotherapy-induced peripheral neuropathy (CIPN), a highly debilitating complication of oncological treatment [5,6]. Hence, diagnosing and treating vitamin B12 deficiency is crucial.

The definition of vitamin B12 deficiency is complex. The conventional laboratory cut-off (serum vitamin B12 concentration <148 pmol/L) indicates vitamin B12 deficiency. Meanwhile, a low or borderline serum B12 concentration (often 150–249 pmol/L) without overt clinical symptoms is called subclinical deficiency [7] and may progress to fully symptomatic deficiency if unrecognized. Also, clinical manifestations are crucial to diagnose vitamin B12 deficiency.

Importantly, hypercobalaminemia—elevated serum vitamin B12 levels—is common in cancer patients. It may be asymptomatic or, paradoxically, coexist with functional vitamin B12 deficiency, where cellular utilization is impaired despite elevated vitamin B12 serum concentrations [8].

Currently, additional markers are proposed to describe vitamin B12 status. These can be categorized as directly measuring vitamin B12 in blood, like serum vitamin B12 and holotranscobalamin, and functional markers, methylmalonic acid and homocysteine, which accumulate when vitamin B12 concentration is inadequate [5,6].

In the USA, vitamin B12 deficiency prevalence is around 5% in the general population [9] and increases with age, reaching approximately 12% in elderly patients [10]. Reported deficiency rates in oncological patients range from 6% to 48%, depending on population and age, with the highest prevalence observed in patients with digestive tract cancers and elderly groups [11,12].

Despite the recognized importance of vitamin B12, its role in oncology remains insufficiently studied. Data on the prevalence and clinical impact of its deficiency or excess in cancer patients, its neuroprotective potential against CIPN, and optimal supplementation strategies are limited. This review summarizes current evidence on vitamin B12 status across solid tumors and gastric precancerous conditions, related laboratory biomarkers, and discusses the implications of extensive surgeries on vitamin B12 levels. It also explores emerging links between vitamin B12 and immune response in cancer. A comprehensive search of PubMed, Scopus, and Web of Science included observational and interventional studies on vitamin B12, cancer, and CIPN.

## 2. Vitamin B12 Sources and Physiological Uptake

Vitamin B12 is an essential, water-soluble vitamin synthesized exclusively by certain bacteria and not naturally present in plants. The main dietary sources for humans are animal products such as red meat, dairy, fish, shellfish, and eggs. Some algae (e.g., *Porphyra* sp., laver–nori) contain large amounts of the inactive form of vitamin B12 for mammals (pseudocobalamin) [13]. Once absorbed, vitamin B12 functions as a coenzyme for the methyl transfer reaction for two metabolic enzymes: (i) methionine synthase that converts homocysteine to methionine; and (ii) L-methylmalonyl-coenzyme A mutase in a reaction that converts L-methylmalonyl-coenzyme A (CoA) to succinyl-CoA [1]. Impaired methionine synthase inhibition results in demyelination of peripheral and central neurons, leading to the characteristic neurological manifestations of vitamin B12 deficiency [2,14]. The hematological effect of B12 deficiency is megaloblastic anemia, which results from disruption of DNA synthesis (see Section 5. Clinical Presentation of Vitamin B12 Deficiency).

### 2.1. Absorption from the Gastrointestinal Tract

Three organs are essential for vitamin B12 absorption: the stomach, pancreas, and terminal ileum. Vitamin B12 is ingested either free or in a protein-bound form. Gastric acid facilitates the release of vitamin B12, which, in its free form, binds to haptocorrin (also known as R binder or R factor), a protein produced in the salivary glands. Pancreatic proteolytic enzymes liberate vitamin B12, which then binds to intrinsic factor (IF) secreted by gastric parietal cells, which are only found in the corpus and fundus mucosa of the stomach. This IF-B12 complex is endocytosed by enterocytes in the terminal ileum, while ~1% of vitamin B12 is passively absorbed along the gastrointestinal tract [2]. Once released from enterocytes, vitamin B12 is transported in the bloodstream bound to transcobalamins; as holotranscobalamin (which accounts for 20–30% of total B12 and is available for cellular uptake), and haptocorrin (70–80% of B12, which is unavailable for metabolic purposes) [15,16]. Since passive absorption of vitamin B12 is very low (around 1%), diseases causing impairment of oxyntic gland function or ileal disorders strongly affect the ability to absorb this vitamin.

### 2.2. Recommended Intake and Body Content

The recommended dietary allowance (RDA) for vitamin B12 for adults (both men and women) is 2.4 mcg/day [1]. Estimates of the average total-body vitamin B12 pool stored mainly in the liver in adults range from 0.6 to 3.9 mg, usually between 2 and 3 mg [1]. Although this amount is relatively small, the vitamin is efficiently recycled through enterohepatic circulation, resulting in a slow overall turnover. Consequently, clinical deficiency develops only after prolonged impaired absorption or intake—typically several years (usually 2–4 years) [17]. However, early metabolic changes such as reduced holotranscobalamin (holo-TC) and increased methylmalonic acid (MMA) and homocysteine (tHcy) may appear earlier, reflecting impaired cellular availability, even when total body content remains relatively preserved (subclinical vitamin B12 deficiency).

## 3. Definition and Laboratory Findings in Vitamin B12 Deficiency

Establishing a precise definition of vitamin B12 deficiency is challenging. Commonly, the test used to assess vitamin B12 is the serum vitamin B12 concentration, with a cut-off of <148 pmol/L, but low/marginal status can already give symptoms. Currently available biomarkers proposed to describe vitamin B12 can be categorized as direct and functional. Direct markers are (i) serum vitamin B12 and (ii) holo-TC, which measure circulating vitamin B12 concentrations. The functional markers: MMA and tHcy accumulate in case of inadequate vitamin B12 concentration [7,18,19,20]. The functional measures can be particularly useful for identifying subclinical and functional vitamin B12 deficiencies. Currently, it is proposed to use both types of markers to assess the vitamin B12 status [18], but they are not always performed due to limited access to those assays or a lack of knowledge. In this section, we described the “pros and cons” of each biomarker.

### 3.1. Serum Vitamin B12 Cut-Off Values and Assessment Limitations

The serum vitamin B12 concentration is the most common marker for assessing vitamin B12 deficiency due to its high availability and relatively low cost, but it carries some limitations. As already stated, low vitamin B12 serum concentration is the biochemical confirmation of vitamin B12 deficiency; but serum B12 within normal range or even elevated does not exclude the deficiency [21,22]. This can be due to the patient’s marginal vitamin B12 status, which can already give symptoms, or analytical limitations, such as laboratory errors or altered B12-binding proteins. Total serum B12 is the sum of vitamin B12 bound to transporting proteins: transcobalamin and haptocorrin; therefore, an increased concentration of these proteins (common in cancer patients) leads to high levels of vitamin B12 [23].

A population study conducted in Canada established reference values for vitamin B12 in this population at 230–730 pmol/L (310–988 pg/mL) [24]. Conversely, a Danish study suggests that reference ranges for vitamin B12 vary by age, being 174–533 pmol/L (235–722 pg/mL), 176–651 pmol/L (238–882 pg/mL), and 118–731 pmol/L (160–990 pg/mL) for the age groups of 18–45, 45–68, and 72–82 years, respectively [25]. There are no specific cut-off values for cancer patients or recommendations on when and how to supplement it.

Regarding analytical limitations, vitamin B12 is commonly measured using assays that employ competitive binding of serum B12 to intrinsic factor. In these assays, a stronger signal detected by the analyzer corresponds to a lower concentration of vitamin B12 in the sample. Anti-intrinsic factor antibodies, which may be present in patients with pernicious anemia, can result in falsely elevated vitamin B12 measurements; however, reagent manufacturers have developed solutions to minimize such interference [22].

It is crucial to consider the variables in results obtained from different analytical platforms, as immune-based methods employ antibodies of varying specificity, and all methods are fraught with measurement uncertainty [26]. Consequently, reference values proposed by different laboratories may differ, and measuring a patient’s vitamin B12 status should be conducted in the same laboratory using the same method [27]. Furthermore, vitamin B12 measurement is susceptible to assay errors. These errors can arise from antibody-mediated interferences, such as heterophile antibodies, macrocomplexes (macro-B12), or rheumatoid factor. Macro-B12 is a non-active immune complex comprised of vitamin B12 and immunoglobulins directed against B12–binding proteins [28], which is, in fact, a relatively rare phenomenon that may occur in patients treated with vitamin B12 injections [22].

### 3.2. Holo-Transcobalamin as a Marker of Vitamin B12 Deficiency

Since vitamin B12 is transported in the system bound to transport proteins—transcobalamin (forming holo-TC, delivering vitamin B12 to the cells) and haptocorrin (unavailable for metabolic purposes), measuring holo-TC, the so-called “active B12,” could be a more reliable assessment of vitamin B12 concentration. Studies have demonstrated this marker’s slightly higher sensitivity and specificity (77% vs. 73% and 76% vs. 72.4% for holo-TC and serum vitamin B12, respectively) [29].

The tentative reference interval for holo-TC is 40–100 pmol/L, with a cut-off value of <40 pmol/L indicating vitamin B12 deficiency [2]. Nevertheless, in some papers, the cut-off <25 pmol/L is proposed, with uncertain status of holo-TC concentration between 25 and 70 pmol/L, warranting additional tests with functional vitamin B12 markers, like MMA to indicate cellular B12 utilization [30].

The main confounding factors for holo-TC are impaired kidney function and genetic variation in the TCN2 gene [2,31,32]. Additionally, the poor availability of holo-TC measurement remains a significant limitation of its routine use. Despite clinical utility and better sensitivity than serum B12 concentration, holo-TC is an imperfect marker of vitamin B12 deficiency and should be interpreted with caution in patients with impaired renal function, as well as combined with other markers, like MMA, in cases of marginal status and suspicion of vitamin B12 deficiency.

### 3.3. Methylmalonic Acid and Homocysteine as Indirect Markers of Vitamin B12 Deficiency

To better reflect vitamin B12 tissue concentration, assessing functional biomarkers, such as MMA and, to a lesser extent, tHcy, is recommended [33]. Both markers accumulate in the serum in cases of vitamin B12 deficiency. Abildgaard et al. demonstrated that MMA levels in healthy adults ranged from 0.12 to 0.40 µmol/L (<68 years old) and from 0.12 to 0.46 µmol/L in the elderly [25]. For MMA, a cut-off of >0.45 µmol/L indicates probable metabolic vitamin B12 deficiency, while >0.75 µmol/L signifies metabolic vitamin B12 deficiency in subjects with normal renal function [29]. MMA measurement requires liquid chromatography with tandem mass spectrometry (LC-MS/MS), available only in specialized laboratories [25], affecting its accessibility. MMA accumulates in case of vitamin B12 deficiency and decreased glomerular filtration rate (GFR) [2,34].

Homocysteine levels vary with age and sex and may rise not only in B12 deficiency but also due to folate or vitamin B6 deficiency, thyroid disorders, smoking, or reduced GFR [2,35] In men, normal tHcy concentrations range from 3.6–10.6 µmol/L at ages 13–25 and 5.2–14.1 µmol/L at ages 26–79. Women generally have lower values, ranging from 2.9–9.5 µmol/L at ages 13–39 and 3.7–10.9 µmol/L at ages 40–79 [24]. Hyperhomocysteinemia, often indicative of vitamin B12 deficiency, is typically defined as tHcy > 15 µmol/L [34].

## 4. Causes of Vitamin B12 Deficiency

Vitamin B12 deficiency among cancer patients can be multifactorial. The deficiency can occur due to (i) cancer site, especially if it involves organs essential for vitamin B12 absorption, such as the stomach, pancreas, or terminal ileum; (ii) undergoing gastrointestinal surgery; and (iii) age above 65 years and associated comorbidities [12]. Importantly, malnutrition, advanced malignancy, and low performance status can lead to vitamin B12 deficiency or hypercobalaminemia [12]. Nonetheless, given its clinical relevance, it should not be overlooked and deserves appropriate evaluation. The possible causes of vitamin B12 deficiency are listed in Table 1, with a more detailed description of the deficiency causes relevant to cancer patients described below.

### 4.1. Causes Related to the Terminal Ileum

As for the ileum, it is the “length that matters”. Studies showed that patients after resection of more than 20 cm of the ileum have significantly lower vitamin B12 levels during the 6-month follow-up compared to baseline. Up to 50% of patients after resection had an abnormal Schilling test result, indicating vitamin B12 malabsorption [45,46]. In patients undergoing right colectomy, extended right colectomy, and ileocecal resection, there was a significant decrease in vitamin B12 concentration over 3–6 months. However, median values of cobalamin stayed within the normal range [45]. Reduced vitamin B_12_ levels following colectomy may be attributed to alterations in the ileal microbiota, particularly the overgrowth of anaerobic bacteria (Small Intestine Bacterial Overgrowth, SIBO), which consume vitamin B12 and impair its absorption [44,47].

### 4.2. Causes Related to the Pancreas

Pancreatic exocrine insufficiency may lead to vitamin B12 malabsorption due to the absence of pancreatic proteases necessary for dissociating vitamin B12 from the vitamin B12-haptocorrin complex. A study of the nutritional status of patients with pancreatic cancer after pancreatoduodenectomy revealed no vitamin B12 deficiency in this population [48]. A study by Osmola et al. revealed that 8% of pancreatic cancer patients initiating systemic chemotherapy have vitamin B12 deficiency [49]. Presented studies have limitations of short observation time after the procedure, and due to ample vitamin B12 body content, it is insufficient to develop a vitamin B12 deficiency.

### 4.3. Vitamin B12 Deficiency Related to the Stomach

#### 4.3.1. Gastric Precancerous Conditions: Helicobacter Pylori-Related Gastritis and Autoimmune Gastritis

Non-cardia intestinal-type gastric cancer develops through a series of changes in the gastric mucosa, known as Correa’s cascade (a stepwise progression from non-atrophic gastritis, usually induced by a chronic *H. pylori* infection, through chronic atrophic gastritis, intestinal metaplasia, and dysplasia, eventually leading to gastric cancer) [50,51,52]. Vitamin B12 deficiency in *H. pylori* gastritis is relatively rare in Occidental countries (<5% of patients) [53]. Recent studies have shown that vitamin B12 levels tend to be lower in *H. pylori*-positive individuals compared to those without the infection, and the eradication of *H. pylori* can lead to improvements in serum vitamin B12 levels, particularly among children [54]. Current guidelines for managing *H. pylori* infection recommend *H. pylori* eradication for patients with vitamin B12 deficiency [55].

Another gastric precancerous condition, autoimmune gastritis (AIG), is a chronic disease in which the immune system attacks parietal cells in the stomach corpus, responsible for gastric acid and intrinsic factor production. This leads to hypochlorhydria, hypergastrinemia, enterochromaffin-like (ECL) cell hyperplasia, and malabsorption of nutrients, especially iron and vitamin B12, resulting in pernicious anemia [53,56,57,58]. Around half of AIG patients are anemic, and even more present with iron and vitamin B12 deficiencies [56]. Since AIG can lead to the development of neuroendocrine tumors of the stomach, and possibly to gastric cancer, patients with AIG and those malignancies should be closely monitored for vitamin B12 deficiency.

#### 4.3.2. Gastric Surgery

The absorption and metabolism of vitamin B12 depend on intrinsic factor produced by the stomach’s parietal cells. Following gastrectomy, intrinsic factor is no longer available, preventing the receptor-mediated endocytosis of vitamin B12 by enterocytes in the terminal ileum. Nonetheless, approximately 1% of vitamin B12 can still be absorbed passively along the gastrointestinal tract.

A meta-analysis found that the prevalence of vitamin B12 deficiency after gastrectomy in patients with gastric cancer was 48.8% [11]. Another study showed cumulative vitamin B12 deficiency rates were 100% after total gastrectomy, with the median time of 15 months to develop this deficiency, and 15.7% for subtotal gastrectomy 4 years after surgery [59]. Patients who are at higher risk of developing anemia after gastrectomy are: women, those after total gastrectomy, and those with diabetes as a comorbidity, whereas a higher postoperative BMI decreased the risk of anemia [60]

Patients who undergo total gastric resection will require lifelong vitamin B12 supplementation. Data show that oral supplementation can be an alternative to parenteral supplementation [44,45]. Eventually, preventive supplementation can be started after the surgery (see Section 7.4 Deficiency vs. preventive supplementation).

## 5. Clinical Presentation of Vitamin B12 Deficiency

Vitamin B12 deficiency can manifest as general, hematological, neuropsychiatric, and gastrointestinal symptoms, individually or in combination. In some cases, it may also be asymptomatic. In cancer patients, hematological abnormalities are particularly significant, as they can contribute to anemia, potentially leading to treatment delays or dose reductions in oncological therapy. Moreover, vitamin B12 deficiency–related neurological symptoms may exacerbate CIPN, resulting in increased patient discomfort.

### 5.1. Hematological Manifestation

The hallmark of vitamin B12 deficiency is macrocytosis (~50% of patients), defined as mean corpuscular volume (MCV) greater than 100 fL with or without anemia (defined as hemoglobin level < 12 g/dL for cancer patients) [61]. In some cases, macrocytosis precedes the development of anemia by months [62]. Decreased reticulocyte count can also be noted. Besides erythrocytes, other cell lines may be affected; in white blood cells, hyper-segmentation of the nuclei in neutrophils may be present, and in the case of an impairment of all marrow precursors, vitamin B12 deficiency may lead to pancytopenia (see Table 2) [61].

### 5.2. Neurological Presentation

Recognizing neurological symptoms associated with vitamin B12 deficiency is critical, as delayed treatment may result in irreversible neurological damage. The symptoms include paraesthesia, while progressive degeneration of the posterior and lateral spinal cords leads to altered proprioception and vibration sensitivity (see Table 2) [63,64]. Importantly, in oncological practice, the neurological symptoms, particularly sensory and motoric neuropathy associated with vitamin B12 deficiency, are similar to those of CIPN. Therefore, assessing vitamin B12 in case of those manifestations is essential since, in B12 deficiency, the supplementation can alleviate symptoms [65] (see Section 9. Vitamin B12 in the Treatment of Chemotherapy-Induced Peripheral Neuropathy). Of note, neurological symptoms in 1/3 of cases preceded haematological symptoms [63,64,66].

**Table 2 nutrients-17-03272-t002:** Clinical manifestations and complete blood count abnormalities in patients with vitamin B12 deficiency.

Symptom	Frequency
**Hematological manifestations** [61]	
Macrocytosis; anemia; hyper-segmented neutrophils	>30%
Leukopenia; thrombopenia	10–15%
Pancytopenia, thrombotic microangiopathy, hemolytic anemia	<5%
Decreased reticulocyte count	unknown
**Neurological symptoms [63]**	
Impaired vibration sense	22%
Paresthesia	20%
Impaired sensory loss	12–15.6%
Ataxia	12%
Memory loss	9%
Psychiatric disorders	5%
**Gastrointestinal symptoms [67]**	
Epigastric pain, heartburn, early satiety	17–35%
Weigh loss	28%
Nausea, diarrhea, vomiting	9–22%
Glossitis	2%
**Thrombo-embolic events**	first manifestation in 20% of cases
**General symptoms**	
Fatigue	not assessed

### 5.3. Gastrointestinal and Other Manifestations

Gastrointestinal symptoms often occur in patients with vitamin B12 deficiency related to AIG. The most often reported symptoms are epigastric pain, early satiety, heartburn, and weight loss. The most characteristic symptom of vitamin B12 deficiency, glossitis, occurs in only 2% of patients [67,68].

Patients with vitamin B12 deficiency have an increased risk of thromboembolic events, possibly due to elevated levels of homocysteine associated with the condition [69,70]. Interestingly, in 20% of cases, thromboembolic events are the sole or inaugural symptom of vitamin B12 deficiency [70] (see Table 2).

## 6. Prevalence of Vitamin B12 Deficiency in Patients with Solid Tumors

The prevalence of vitamin B12 deficiency among cancer patients depends on several factors. Aside from the cancer site (whether organs involved in vitamin B12 absorption are affected or not) and the treatments associated with cancer, which may differ between cancer types, the definition of vitamin B12 deficiency employed in the study (whether using a classical cut-off value or functional markers) impacts the prevalence rates. Additionally, the geographical regions described, along with their underlying vitamin B12 deficiency levels in the general population, translate into the percentage of patients entering oncological treatment with deficiency. Most studies cited below used the classical definition of vitamin B12 deficiency (serum vitamin B12 below the normal range, unless otherwise stated).

Regarding the prevalence of vitamin B12 deficiency in patients with gastrointestinal cancers, 48.8% of patients with gastric cancer after gastrectomy were affected, as already mentioned in Section 5 [11]. Vitamin B12 deficiency can be anticipated in 18% of patients after esophagectomy due to cancer (in this study, serum vitamin B12, MMA and holo-TC were used as markers) [71]. Another study found that vitamin B12 was lower in patients with gastric or colon cancer compared to other cancer types [12].

Concerning the vitamin B12 status in patients with organs not involved in vitamin B12 absorption as the primary cancer site: the concentration of vitamin B12 in melanoma patients was comparable to that of healthy controls [72], in patients with gynecological cancers during oncological treatment vitamin B12 deficiency was estimated at 5% [73] Patients with breast cancer were characterized by the highest median B12 value among all cancer types in hospitalized cancer patients [12]. Vitamin B12 deficiency (vitamin B12 < 200 pg/mL) in hospitalized cancer patients in Italy was present in 14% of patients, whereas low vitamin B12 (200–299 pg/mL), which can already be symptomatic, was present in 19.4% of patients, and it was more common in early-stage cancer and elderly patients [12].

In a study from Turkey evaluating patients with newly diagnosed cancer, with a predominance of breast and colorectal cancer patients in the cohort, it was found that 39.3% of the patients were diagnosed as cobalamin-deficient, compared to 18.9% of controls without cancer [74]. Similarly, a study from India assessing treatment-naïve cancer patients showed that 47% of anemic participants were vitamin B12-deficient [75]. The latter two studies show differences in the prevalence of vitamin B12 deficiency in different geographical regions, where a higher prevalence of vitamin B12 in the general population translates into a higher number of cancer patients who are vitamin B12 deficient. In summary, vitamin B12 deficiency is estimated at 6–48% in cancer patients, is higher in patients with digestive tract cancers, as well as in early-stage cancer and the elderly, and depends on the studied population.

### 6.1. Pediatric Onco-Hematological Patients

The prevalence of vitamin B12 deficiency in children in the USA is around 5% [9], while in Turkey, 4.5%, with 25% of children having borderline B12 levels [76]. Regarding oncolohematological patients, studies performed in India and Turkey showed that vitamin B12 deficiency was present in 8–40% of patients pre-treatment [77,78,79], whereas, during the maintenance therapy for leukemia, 8% had vitamin B12 deficiency [80]. The study performed in India showed that the prevalence of vitamin B12 deficiency in pediatric patients is similar in solid and hematological malignancies and occurs almost as often as iron deficiency [78]. Data regarding the prevalence of vitamin B12 deficiency in Europe and the USA in oncohematological pediatric patients are lacking, probably because this condition is properly screened and treated in this group.

### 6.2. Elderly Patients

Limited data exist regarding the prevalence of vitamin B12 deficiency in oncogeriatric patients. However, among the elderly, this deficiency is more common than in younger adults, with an estimated prevalence of 11–14% [81,82]. The causes of this deficiency in the elderly are food-cobalamin malabsorption in 2/3 of cases, and autoimmune gastritis in 1/3 of cases [10]. Another study showed that oncological patients >65 years old are more susceptible to present with vitamin B12 deficiency [12]. Given these findings, screening for vitamin B12 deficiency in oncogeriatric patients is particularly important; studies assessing the exact prevalence of vitamin B12 deficiency in the elderly are needed.

## 7. Clinical Management of Vitamin B12 Deficiency

Data on the clinical management of vitamin B12 deficiency in cancer patients are scarce. There are no clinical trials or specific guidelines for vitamin B12 supplementation in cancer patients. However, since the management should not differ significantly from the general population, this chapter includes studies on supplementation principles regardless of the deficiency etiology.

### 7.1. Forms of Vitamin B12

Different chemical forms of vitamin B12 are available for parenteral or oral administration, such as cyanocobalamin, hydroxocobalamin, methylcobalamin, and adenosylcobalamin. Cyanocobalamin is the usual form of supplementation. Hydroxycobalamin is used in some countries for B12 supplementation and as a cyanide poisoning antidote; its biological availability is higher and retained in the body longer than cyanocobalamin [83].

### 7.2. Route of Administration

The two main routes for supplementing vitamin B12 deficiency are the parenteral route, which most commonly uses the intramuscular (IM) method, and the subcutaneous (SQ) route. The second group includes the oral (PO), sublingual (SL), and intranasal (IN) routes, with the oral route being the most frequently utilized for supplementation.

Passive enteral absorption of vitamin B12 is approximately 1% of any oral dose, with interindividual variability in this vitamin’s absorption [84]. The efficacy of various routes of administration has been compared in several studies and meta-analyses. The latest Cochrane review by Wang et al. assesses the effects of daily PO vs. IM supplementation of vitamin B12 in patients with known deficiency [85]. When administered at the daily dose of 1000 µg for oral and a monthly dose of 1000 µg for intramuscular supplementation, there was no significant difference in the increase of vitamin B12 blood levels. One of the studies used a dose of 2000 µg for oral administration, resulting in higher blood levels than intramuscular. The systematic review of Butler et al. concluded that a daily dose of 2000 µg of oral vitamin B12 and a maintenance dose of 1000 µg weekly were as effective as intramuscular administration, inducing a short-term response in the symptoms of deficient patients [86]. The latest systematic review by Abdelwahab et al., which included 13 studies comparing the PO, SL, and IM routes of administration in deficient patients, reveals no statistical difference among these routes in their ability to increase vitamin B12 levels [87]. In the largest randomized trial, oral vitamin B12 (1 mg/day for 8 weeks, then 1 mg/week for 1 year) was compared with intramuscular administration (1 mg every other day for 8 weeks, then monthly for 1 year). Oral supplementation proved non-inferior at week 8, with only marginal differences observed at week [88]. In a retrospective analysis of over 4000 patients, Bensky et al. found that SL vitamin B12 supplementation was associated with better outcomes than the IM group [89].

### 7.3. Vitamin B12 Supplementation Dosage

Data on optimal vitamin B12 dosing regimens are limited. Treatment usually begins with intramuscular loading doses for 10–14 days, or 1000 µg orally per day (up to 2000 µg in some studies) until symptom resolution. Maintenance therapy typically consists of monthly IM or weekly oral doses. Regular monitoring—particularly with oral therapy—is recommended, with a switch to IM administration if response is inadequate. Lifelong supplementation is required in irreversible malabsorption (e.g., after gastrectomy or ileal resection).

### 7.4. Deficiency Versus Preventive Supplementation

In the previous paragraph, we addressed supplementation regimens for vitamin B12 deficiency with or without symptoms. In situations of high risk of deficiency, such as total gastrectomy or ileal resection >20 cm, the supplementation regimen (oral cyanocobalamin 1000 µg/week) should be started soon after the surgery, despite the possibility that vitamin B12 levels could remain within the normal range for several months up to years after the procedure.

### 7.5. Monitoring of Supplementation Effectiveness

Despite its importance, monitoring the therapeutic response to cobalamin deficiency treatment is often overlooked. A full hematologic response confirms true cobalamin deficiency, while an attenuated response may indicate alternative diagnoses or complicating factors [62].

The earliest objective marker of treatment’s response is a peak in reticulocyte count, expected approximately one week after initiating therapy. The magnitude of response correlates with the severity of anemia [62].

The definitive hematologic milestone is normalization of the complete blood count and MCV after a few weeks of treatment. Failure of homocysteine or MMA levels to normalize within the first week raises suspicion of a misdiagnosis. In contrast, serum cobalamin and holo-TC concentrations offer limited utility during follow-up, as they rise non-specifically with supplementation [62].

### 7.6. Vitamin B12 Supplementation and Increased Cancer Risk

Patients with cancer have been reported to show elevated plasma concentrations of vitamin B12 (see below), thus causing uncertainties regarding safety of vitamin B12 supplementation. A recent systematic review summarizing available data did not show evidence to assume that high plasma vitamin B12, high B12 intake, or treatment with vitamin B12, is causally related to cancer [90].

## 8. Hypercobalaminemia

Hypercobalaminemia, defined as vitamin B12 levels above 950 pg/mL (700 pmol/L) [8], is often observed in cancer patients, but the exact mechanism and clinical implications remain elusive. Clinically, it can be asymptomatic or paradoxically accompanied by signs of deficiency, reflecting a functional deficiency linked to defects in the tissue uptake of vitamin B12 [8].

The prevalence of high B12 is present in 13–14% of hospitalized adult patients [23,31] and 17% of hospitalized cancer patients [12]. In this chapter, we try to explain the phenomenon of hypercobalaminemia concerning cancer patients.

### 8.1. Underlying Mechanisms

Several mechanisms can lead to high B12 levels: (i) excessive intake or administration, (ii) release from reservoirs (such as from the liver in cases of hepatic insufficiency and/or metastases), (iii) decreased vitamin B12 clearance, in cases of renal or hepatic insufficiency, (iv) an increase in vitamin B12 transporting proteins (transcobalamin or haptocorrin) produced by cancer cells, and (v) immune complexes, where vitamin B12 bound to immunoglobulins can falsely elevate measured levels [1,87].

Incidental findings of high vitamin B12 should always be followed by additional investigation. In a study of 4800 individuals, of which 250 had high B12 levels, structural liver disease was detected in 23.6%, solid cancer in 18.2%, and 7.1% had malignant hemopathy with a median time to diagnosis of 10 months [91]. Hypercobalaminemia is often the first laboratory finding of oncological disease [23], and is linked with higher cancer risk [92,93]. Another study revealed that high cobalamin levels (>1000 ng/L) are associated with an increased risk of cancer, also after adjustment for other causes of hypercobalaminemia, i.e., liver diseases, chronic kidney failure, autoimmune or inflammatory diseases, and excessive B12 supplementation [94]. More recent data from linked elevated vitamin B12 levels exceeding 1000 ng/mL with the occurrence of solid cancer, but only in cases of hypercobalaminemia persistent for over a month [95].

Regarding oncological patients, high vitamin B12 concentrations are often seen in patients with hematological disorders, especially in chronic myelogenous leukemia and hypereosinophilic syndrome [96]. High vitamin B12 levels are seen in ~50% of the patients with hepatocellular carcinoma [97] and in patients with liver metastases [98], with a possible underlying mechanism of decreased cobalamin–haptocorrin clearance and cobalamin release during hepatic cytolysis caused by cancer cells [99].

### 8.2. Hypercobalaminemia as a Prognostic Factor in Palliative Medicine

In recent years, high vitamin B12 levels have been recognized as an adverse prognostic factor in several observational studies. Elevated vitamin B12 level was proven to be an independent predictive factor for mortality in cancer patients in the palliative care setting [100]. In another study, a vitamin B12/C-reactive protein Index (BCI) higher than 400.000 could discriminate the group of patients with the lowest median survival (median 29 days) [101]. In a study assessing oncogeriatric patients, BCI was a prognostic factor of mortality within 3 months and unplanned hospitalizations [102].

## 9. Vitamin B12 in the Treatment of Chemotherapy-Induced Peripheral Neuropathy

Chemotherapy-induced peripheral neuropathy is a common side effect of various cytotoxic drugs. The incidence of CIPN of any grade varies depending on the cytotoxic drugs and can be as high as 85% for oxaliplatin. CIPN symptoms are predominantly sensory; in more advanced cases of CIPN, motoric involvement can occur, often affecting hands and feet in a characteristic “glove-and-sock” pattern [103]. These symptoms are similar to those associated with vitamin B12 deficiency, and this deficiency aggravates CIPN [5]. To date, there are no recommendations regarding pharmacological prophylaxis of CIPN. Cooling and compression of the extremities during chemotherapy infusion was recently confirmed to reduce CIPN symptoms during the treatment with paclitaxel [104]. For the CIPN treatment, only oral duloxetine is recommended as its efficacy was proven in a randomized controlled trial [105]. Because vitamin B12 deficiency is known to have neurological manifestations, medical oncologists often prescribe B vitamin supplementation in cases of CIPN. However, the evidence supporting its use in this scenario is scarce and is listed in Table 3.

Schloss and colleagues conducted the only randomized placebo-controlled trial assessing the efficacy of B vitamin (including vitamin B12) administration in CIPN prophylaxis. While a trend was noted, especially concerning reducing both the onset and severity of CIPN, no statistical significance was detected in the Total Neuropathy Score (TNS). However, patients reported a significant reduction in sensory symptoms of CIPN when receiving the B vitamin complex [106]. In a retrospective study assessing functional vitamin B12 deficiency (increased/normal vitamin B12 concentration but elevated MMA and/or tHcy) in patients with advanced cancer, three patients with increased MMA values and neuropathy were treated with cyanocobalamin. In all three of them, neurological symptoms were improved [65].

Overall, there is limited clinical evidence on using vitamin B12 in patients with CIPN as a treatment or for prophylaxis, particularly in the absence of vitamin B12 deficiency. According to the current European Society of Medical Oncology (ESMO) guidelines for CIPN, treatment with B vitamins cannot be recommended in this setting [107]. This topic requires further clinical research.

## 10. Vitamin B12 and Immune Response in Cancer Treatment

The possible link between Vitamin B12 and immune response, and in particular response to immunotherapy in cancer, has been a subject of different experimental and clinical studies with conflicting results. Indeed, vitamin B12 is essential for blood cell production, and thus its deficiency may lead to pancytopenia, including leucopenia and impaired immunity. It has also been shown that vitamin B12 may have immunomodulating effects in cancer through tumor-specific properties of the vitamin B12 receptor [108]. Some experimental studies using a mouse model of pancreatic adenocarcinoma showed that vitamin B12 may act as a leucine-rich repeat kinase 2 (LRRK2) inhibitor and sensitize cancer cells to immunotherapy with anti-PD-L1 agents. The PD-L1 blockade combined with LRRK2 inhibition could be a novel potential strategy for pancreatic adenocarcinoma [109].

On the other hand, hypervitaminosis B12 has been shown to be a negative predictor of immunotherapy outcomes in cancer patients treated with immune checkpoint inhibitors [103]. Altogether, the scarce data available so far indicate that there is a logical indication to maintain a correct vitamin B12 level in cancer patients. Both deficiency and excess may be detrimental, albeit via different mechanisms. However, the precise immunological roles of B12 in cancer patients remain insufficiently characterized, and further research—especially well-controlled clinical trials—is necessary to determine whether active correction or modulation of B12 levels could improve immunotherapy outcomes.

## 11. Conclusions

Abnormal vitamin B12 status—both deficiency and excess—is common among cancer patients. Reported deficiency rates range from 6% to 48%, depending on population and age, with the highest prevalence observed in patients with digestive tract cancers and elderly groups. Hypercobalaminemia occurs in approximately 17% of cancer patients and, in the general population, may serve as a marker of malignancy. Currently, no randomized controlled trials have evaluated the efficacy or safety of vitamin B12 supplementation in cancer patients; thus, deficiency should be managed according to general population guidelines. Although the role of vitamin B12 in preventing or treating CIPN remains uncertain, patients presenting with neuropathic symptoms should be assessed for possible B12 deficiency, as it may exacerbate this condition.

## Figures and Tables

**Table 1 nutrients-17-03272-t001:** Causes of vitamin B12 deficiency in the general population.

Causes of B12 Deficiency
**Decreased vitamin B12 supply**
Vegetarian and vegan diet [36]
Malnutrition
Alcoholism
**Related to the stomach**
Autoimmune gastritis (lack of intrinsic factor)
Gastric surgery, total gastrectomy
Bariatric surgery, Roux-en-Y gastric bypass, and sleeve gastrectomy
**Pancreatic insufficiency**
Chronic pancreatitis
Pancreatectomy
**Ileal**
Resection of terminal ileum (>20 cm)
Small Intestine Bacterial Overgrowth (SIBO) *
Inflammatory Bowel Disease (particularly Crohn’s Disease)—can cause the ileum’s “functional” impairment [37,38]
**Genetic** [39]
Imerslund–Gräscbeck syndrome
Congenital Intrinsic Factor deficiency (genetic syndrome leading to vitamin B12 deficiency usually manifest itself at the early age)
**Drug-related malabsorption**
Chronic proton pump inhibitor usage [40] and/or H2-recepter antagonist due to subsequent hypochlorhydria
Metformin usage (the mechanism remains unknown, but probably due to altered B12-intrinstic factor interactions) [41]
Colchicine [42]
Nitrous oxide (N_2_O), known also as a laughing gas, frequent consumption [43]
**Other causes**
Older age and related malabsorption

* In SIBO, it is important to carry out vitamin B12 supplementation parenterally because the bacteria in the intestine’s lumen increase consumption [44].

**Table 3 nutrients-17-03272-t003:** Studies on vitamin B12 supplementation in patients with chemotherapy-induced peripheral neuropathy (CIPN).

Author, Year	Type of Study	Intervention	Chemotherapy	Result
Schloss JM, Colosimo M, et al., 2015 [5]	Case study	Vitamin B12 intramuscular	Paclitaxel for breast cancer	Vitamin B12 i.m and vitamin B complex **improved** CIPN associated with B12 deficiency and paclitaxel chemotherapy in a patient with breast cancer
Solomon L., 2016 [65]	Retrospective study	Vitamin B12 therapy (5 patients oral supplementation, 1 patient—intramuscular)	No data	From 241 cancer subjects, 3 patients with both functional vitamin B12 deficiency and neurologic abnormalities had **clinical improvement** after B12 therapy
Schloss JM, Colosimo M, et al., 2017 [106]	Randomized Clinical Trial	Vitamin B group oral administration vs. placebo	Taxanes, oxaliplatin or vincristine for various neoplasms	Patients with an oral B group vitamin supplementation **did not have superior results** to placebo group for the prevention of CIPN measured with total neuropathy score. **Patients** taking the B group vitamin **perceived a reduction in sensory peripheral neuropathy** in the Patient Neurotoxicity Questionnaire

## Data Availability

No new data were created.

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
