# Peer review of "Vitamin B12 in Cancer Patients: Clinical Insights into Deficiency, Excess, Diagnosis, and Management"

_nutrients, 2025, doi:10.3390/nu17203272_

Round 1
Reviewer 1 Report
Comments and Suggestions for Authors
A paper covering the title would be great.
However, the present paper represents a broad and at times undigested and misleading description of the field of so-called B12 deficiency with an insufficient summary of data relating to the field of B12 and cancer. This is reflected also in the conclusion, that does not give any new information of relevance.
Below are some points to consider.
- The term B12 deficiency should be clarified. How is it defined – and is it defined in the same manner in all the papers cited?
An example: First few lines in the introduction apparently base its data on “B12 deficiency” simply on laboratory values (ref 2). It should be clarified and stated whether B12 deficiency simply refers to B12 values above or below a certain value. - Please explain the meaning of CIPN. Currently you need to go to section 9 to have an explanation.
- The sentence: “As a result, cobalamin deficiency can lead to hematologic and neurologic symptoms [1].” Needs explanation. As a result of what?
- In the sentence “…vitamin B12 is transported into the bloodstream” It should be … transported in – not into.
- The section of intake and storage is unclear and should be rewritten referring to body content rather than body stores. Biomarkers of B12 starts to decline soon after an impaired uptake. The old concept of “B12 store” is likely based on the very slow turnaround time for B12 combined with finding of overt symptoms, only once total body content of B12 has declined markedly.
- Section 3 is unclear. It needs an introduction outlining that B12 deficiency is NOT a biochemical diagnosis, and then briefly how you intend to use the markers that may support or discharge a clinical B12 deficiency. The section of the individual biomarkers should be abbreviated and focused. Some suggestions are given below.
Consider starting by an introductions to the biomarkers employed, and how you plan to use the biomarkers and then make a section on each of them. - The section on holoTC needs rewriting. It is repetitive and sketchy, not the least concerning the role of the kidney and its superiority in relation to total B12. Its emphasis as to a role in individuals with a possible increase in haptocorrin is to the point.
- The introduction to section 3.3 is unclear. What do you mean by the first sentence?
- Please note that B12 measures are NOT characterized as immune assays, and rarely even involve an antibody in the assay. Common for the assays is the use of intrinsic factor where added labeled B12 competes with B12 in the sample for binding to intrinsic factor.
- The description of macro B12 is misleading. Unexpected high levels of B122 is caused by an increase in haptocorrin and/or transcobalamin, and only rarely include the presence of gammaglobulin. PEG precipitation is nonspecific and should not be used.
- Section 4 should be rewritten and markedly abbreviated. The table seems nicely to summarize causes, so that all that is needed in the text may be reflections as to which of those causes are of major importance in cancer patients.
- Table 2 would increase in value if it is focused on the theme of the paper, patients with malignancy. The term laboratory findings seems misleading since you do not include most of the laboratory findings.
- Section 6. Please be clear as what you mean by vitamin B12 deficiency. Is it based on a simple blood test, and should really be a plasma B12 below a certain value??
The relevant thing to present is, whether to expect a higher frequency of B12 deficiency in cancer patients. In India a fair amount of the population has an inadequate intake of B12, and would be classified as deficient based on that. - 1. You can not compare data from US with Turkey and India. The description of cause seems speculative. Have you looked for whether general poor nutrition plays a role.
- Section 7 seems to be an undigested summary of previous studies and is more confusing than helpful. One would expect to get information such as cancer patients with B12 deficiency should be treated like other individuals with this condition, in brief this includes….. In addition, the following should be considered especially in patients with cancer.
The section should be abbreviated and focused. - Section 8 seems central for this paper. However, it is again very superficial and adds little new. Example: what do you mean by “Recent data show that high vitamin B12 levels are underestimated and rather frequent anomaly”.
- 1 is repetitive and incorrect. The most common cause for an increased plasma B12 not related to intake is an increased level of haptocorrin. This is a likely explanation for the increased level in many cancer patients. In the paper only a decreased clearance of haptocorrin is mentioned. This should be amended. The statement “deficiency in the affinity of transcobalamin for vitamin B12” does not make sense. In addition, the sentence: “also after adjustment for other causes of hypercobalaminemia, i.e., myeloid blood malignancies” does not make sense. This is also a malignant disease.
- Section 9 and 10 are potentially important but would need a more in depth and critical analysis of existing data.
Author Response
- ReThe term B12 deficiency should be clarified. How is it defined – and is it defined in the same manner in all the papers cited?
An example: First few lines in the introduction apparently base its data on “B12 deficiency” simply on laboratory values (ref 2). It should be clarified and stated whether B12 deficiency simply refers to B12 values above or below a certain value.
Response: Thank you for the valuable remark, we provided the definition of vitamin B12 and described its complexity: “
The definition of vitamin B12 deficiency is complex. The conventional laboratory cut-off (serum vitamin B12 concentration <148 pmol/L) indicates a vitamin B12 deficiency. Meanwhile, a low or borderline serum B12 concentration (often 150–249 pmol/L) without overt clinical symptoms is called subclinical deficiency [7] and may progress to fully symptomatic deficiency if unrecognized. Also, clinical manifestations are crucial to diagnose vitamin B12 deficiency. [...]"
- Comment: Please explain the meaning of CIPN. Currently you need to go to section 9 to have an explanation.
Response: Thank you for this remark. We explained this abbreviation in the text.
- The sentence: “As a result, cobalamin deficiency can lead to hematologic and neurologic symptoms [1].” Needs explanation. As a result of what?
Response: Thank you for this remark, as the message coming from this sentence is not clear. We modified it to: “Impaired methionine synthase inhibition results in demyelination of peripheral and central neurons, leading to the characteristic neurological manifestations of vitamin B12 deficiency [2,8]. The haematological effect of B12 deficiency is megaloblastic anaemia, which results from disruption of DNA synthesis (– see Chapter 5. Clinical Presentation of Vitamin B12 Deficiency).
- Comment: In the sentence “…vitamin B12 is transported into the bloodstream” It should be … transported in – not into.
Response: Thank you for the comment. We corrected it following your advice. “Once released from enterocytes, vitamin B12 is transported in the bloodstream…”
- The section of intake and storage is unclear and should be rewritten referring to body content rather than body stores. Biomarkers of B12 starts to decline soon after an impaired uptake. The old concept of “B12 store” is likely based on the very slow turnaround time for B12 combined with finding of overt symptoms, only once total body content of B12 has declined markedly.
Response: We thank the reviewer for this important clarification. We agree that the traditional concept of “vitamin B12 stores” is outdated and does not accurately reflect the continuous turnover and redistribution of vitamin B12 within the body. We have therefore revised the section to refer to total body content, emphasizing that functional biomarkers of B12 status with “modern” markers like holotranscobalamin, methylmalonic acid may decline soon after absorption is impaired, even before total body content markedly decreases causing true vitamin b12 deficiency. The text was rewritten accordingly for clarity and accuracy: “The recommended dietary allowance (RDA) for vitamin B12 for adults (both men and women) is 2.4 mcg/day [10]. Estimates of the average total-body vitamin B12 pool stored mainly in the liver in adults range from 0.6 to 3.9 mg, usually between 2 and 3 mg [10]. Although this amount is relatively small, the vitamin is efficiently recycled through enterohepatic circulation, resulting in a slow overall turnover. Consequently, clinical deficiency develops only after prolonged impaired absorption or intake—typically several years (usually 2–4 years) [11]. However, early metabolic changes such as reduced holotranscobalamin and increased methylmalonic acid and homocysteine may appear earlier, reflecting impaired cellular availability, even when total body content remains relatively preserved (subclinical vitamin B12 deficiency).
- Section 3 is unclear. It needs an introduction outlining that B12 deficiency is NOT a biochemical diagnosis, and then briefly how you intend to use the markers that may support or discharge a clinical B12 deficiency. The section of the individual biomarkers should be abbreviated and focused. Some suggestions are given below.
Consider starting by an introductions to the biomarkers employed, and how you plan to use the biomarkers and then make a section on each of them.
Response: thank you for this suggestion, we started this section with an introduction: “Establishing a precise definition of vitamin B12 deficiency is challenging. Commonly, the test used to assess vitamin B12 is the serum vitamin B12 concentration, with a cut-off <148 pmol/L, but low/marginal status can already give symptoms. Currently available biomarkers proposed to describe vitamin B12, can be categorized as direct and functional. Direct markers are (i) serum vitamin B12 and (ii) holo-TC, which measure circulating vitamin B-12 concentrations. The functional markers: methylmalonic acid (MMA) and homocysteine (tHcy), accumulate in case of inadequate vitamin B12 concentration [5,6,14,15]. The functional measures can be useful for identifying subclinical vitamin B12 deficiency. Currently, it is proposed to use both types of markers to assess the vitamin B12 status [5], but due to limited access to those assays or a lack of knowledge, they are not always performed. Importantly, clinical manifestations are crucial to diagnose vitamin B12 deficiency. In this section, we described the “pros and cons“ of each biomarker.”
- The section on holoTC needs rewriting. It is repetitive and sketchy, not the least concerning the role of the kidney and its superiority in relation to total B12. Its emphasis as to a role in individuals with a possible increase in haptocorrin is to the point.
Response: Thank you for this remark. Indeed it is very chaotic, we rewrote this section: “Since vitamin B12 is transported in the system bound to transport proteins – transcobalamin (forming holo-TC, delivering vitamin B12 to the cells) and haptocorrin (unavailable for metabolic purposes), measuring holo-TC, the so-called “active B12,” could be a more reliable assessment of vitamin B12 concentration. Studies have demonstrated this marker's slightly higher sensitivity and specificity (77% versus 73% and 76% versus 72.4% for holo-TC and serum vitamin B12, respectively) [24]. The tentative reference interval for holo-TC is 40–100 pmol/L, with a cut-off value of <40 pmol/L indicating vitamin B12 deficiency [2]. Nevertheless, in some papers, the cut-off <25 pmol/L is proposed, with uncertain status of holo-TC concentration between 25 and 70 pmol/L, warranting additional tests with functional vitamin B12 markers, like MMA to indicate cellular B12 utilization [25]. The main confounding factor for holo-TC is impaired kidney function, and genetic variation in the TCN2 gene [2,26,27]. Additionally, the poor availability of holo-TC measurement remains a significant limitation of its routine use. Despite clinical utility and better sensitivity than serum B12 concentration, holo-TC is an imperfect marker of vitamin B12 deficiency and should be interpreted with caution in patients with impaired renal function, as well as combined with other markers, like MMA, in cases of marginal status and suspicion of vitamin B12 deficiency.”
- The introduction to section 3.3 is unclear. What do you mean by the first sentence?
Response: Indeed the message is unclear, therefore we have rewritten the paragraph to make it more concise and erased the first sentence.
- Please note that B12 measures are NOT characterized as immune assays, and rarely even involve an antibody in the assay. Common for the assays is the use of intrinsic factor where added labeled B12 competes with B12 in the sample for binding to intrinsic factor.
Response: Thank you for this important remark, you are absolutely right. It was a simplification that shouldn’t have been used. However, for measuring active vitamin B12 (for example on Roche analyzer) classical immunoassay ECLIA based on streptavidin-biotin complexation is used. It has been corrected: “While low vitamin B12 serum concentration (ie <148 pmol/L) is the biochemical confirmation of vitamin B12 deficiency, concentration of serum vitamin B12 within normal range or even elevated does not exclude the deficiency. This can be due to the patient’s marginal vitamin B12 status, which can already give symptoms, or analytical limitations, such as laboratory errors or altered B12-binding proteins. Total serum B12 is the sum of vitamin B12 bound to transporting proteins: transcobalamin and haptocorrin; therefore, an increased concentration of these proteins (common in cancer patients) leads to high levels of vitamin B12 [22].
Vitamin B12 is commonly measured using assays that employ competitive binding of serum B12 to intrinsic factor. In these assays, a stronger signal detected by the analyzer corresponds to a lower concentration of vitamin B12 in the sample. Anti-intrinsic factor antibodies, which may be present in patients with pernicious anemia, can result in falsely elevated vitamin B12 measurements; however, reagent manufacturers have developed solutions to minimize such interference [23].”
- The description of macro B12 is misleading. Unexpected high levels of B122 is caused by an increase in haptocorrin and/or transcobalamin, and only rarely include the presence of gammaglobulin. PEG precipitation is nonspecific and should not be used.
Response: It has been corrected, to underline that macrocomplexes are not the most common cause of vitamin B12 increase, the PEG precipitation part was erased.:” Macro- B12 is a non-active immune complex comprised of vitamin B12, and immunoglobulins directed against B12–binding proteins [26], which is, in fact relatively rare phenomenon that may occur in patients treated with vitamin B12 injections [23]. [...]"
- Comment: Section 4 should be rewritten and markedly abbreviated. The table seems nicely to summarize causes, so that all that is needed in the text may be reflections as to which of those causes are of major importance in cancer patients.
Response: Thank you for this remark. In this section we have changed the introduction [see below], listed the causes in the Table and shortened the text describing the causes, focusing on those mostly relevant to cancer patients.
“Vitamin B12 deficiency among cancer patients can be multifactorial. The deficiency can occur due to: (i) cancer site, especially if it involves organs essential for vitamin B12 absorption, such as the stomach, pancreas, or terminal ileum; (ii) undergoing gastrointestinal surgery; (iii) age above 65 years and associated comorbidities [27]. Importantly, malnutrition, advanced malignancy, and low performance status can lead to vitamin B12 deficiency or hypercobalaminemia [27]. Nonetheless, given its clinical relevance, it should not be overlooked and deserves appropriate evaluation. The possible causes of vitamin B12 deficiency are listed in Table 1, with a more detailed description of the deficiency causes relevant to cancer patients described below.
- Comment: Table 2 would increase in value if it is focused on the theme of the paper, patients with malignancy.
Response: thank you for this remark. Since there is no such data regarding only cancer patients, we left the table in the previous form, but described the clinical manifestations related to cancer patients in the text:
“Vitamin B12 deficiency can manifest as general, hematological, neuropsychiatric, and gastrointestinal symptoms, individually or in combination. In some cases, it may also be asymptomatic. In cancer patients, hematological abnormalities are particularly significant, as they can contribute to anemia, potentially leading to treatment delays or dose reductions in oncological therapy. Moreover, vitamin B12 deficiency–related neurological symptoms may exacerbate chemotherapy-induced peripheral neuropathy, resulting in increased patient discomfort.” [...]
neurological symptoms: “Importantly, in oncological practice, the neurological symptoms, particularly sensory and motoric neuropathy associated with vitamin B12 deficiency, may resemble those of chemotherapy-induced peripheral neuropathy. Therefore, assessing vitamin B12 in case of those manifestations is essential since, in B12 deficiency, the supplementation can alleviate symptoms [57] (– see Chapter 9. Vitamin B12 in the Treatment of Chemotherapy-Induced Peripheral Neuropathy).
Comment: The term laboratory findings seems misleading since you do not include most of the laboratory findings.
Response: Thank you for this remark, we changed the Table title to “Clinical manifestations and complete blood count abnormalities in patients with vitamin B12 deficiency”, which should be more appropriate.
- Section 6. Please be clear as what you mean by vitamin B12 deficiency. Is it based on a simple blood test, and should really be a plasma B12 below a certain value??
The relevant thing to present is, whether to expect a higher frequency of B12 deficiency in cancer patients. In India a fair amount of the population has an inadequate intake of B12, and would be classified as deficient based on that.
Response: Thank you for this comment: we clarified what was the definition of vitamin B12 deficiency in cited studies and added information that vit B12 deficiency in general population directly translates to the percentage of cancer pateitns affected by it: “ The prevalence of vitamin B12 deficiency among cancer patients depends on few factors, including (i) cancer site (those involved in vitamin B12 absorption versus not involved), (ii) procedures associated with cancer treatment, like extensive surgeries, notably involving gastrointestinal tract, (iii) definition of vitamin B12 deficiency used (classical cut-off value versus functional deficiency, markers used to describe the deficiency like vitamin B12 concentration, MMA/homocysteine, and (iv) geographical region with its underlying vitamin B12 deficiency in the general population, as it translates into the percentage of patients entering oncological treatment with deficiency or not. Most studies cited below used the classical definition of vitamin B12 deficiency (serum vitamin B12 below the normal range, unless otherwise stated).
- You can not compare data from US with Turkey and India. The description of cause seems speculative. Have you looked for whether general poor nutrition plays a role.
Response: You are right that this sentence was unfortunate, it wasn not our goal to compare these data but to show perspectives from different countries. We corrected this paragraph.
- Section 7 seems to be an undigested summary of previous studies and is more confusing than helpful. One would expect to get information such as cancer patients with B12 deficiency should be treated like other individuals with this condition, in brief this includes….. In addition, the following should be considered especially in patients with cancer.
The section should be abbreviated and focused.
Response: Thank you for this remark. Following your advice we shortened this section, where the information as the administration route, dose, supplementation effectiveness and preventive supplementation are briefly described. Since there is no specific data regarding cancer patients, we would refrain from giving “simple guidelines” on this topic.
- Section 8 seems central for this paper. However, it is again very superficial and adds little new. Example: what do you mean by “Recent data show that high vitamin B12 levels are underestimated and rather frequent anomaly”.
Response: Thank you for the comment, indeed this sentence is not clear. We rewrote the introduction to this chapter for clarity: “Hypercobalaminemia, defined as vitamin B12 levels above 950 pg/ml (700 pmol/l) [88], is often observed in cancer patients, but the exact mechanism and clinical implications remain elusive. Clinically, it can be asymptomatic or paradoxically accompanied by signs of deficiency, reflecting a functional deficiency linked to defects in the tissue uptake of vitamin B12 [88].The prevalence of high B12 is present in 13–14% of hospitalized adult patients [89,90] and 17% of hospitalized cancer patients [32]. In this chapter, we try to explain the phenomenon of hypercobalamiemia in relation to cancer patients.’
- 1 is repetitive and incorrect. The most common cause for an increased plasma B12 not related to intake is an increased level of haptocorrin. This is a likely explanation for the increased level in many cancer patients. In the paper only a decreased clearance of haptocorrin is mentioned. This should be amended. The statement “deficiency in the affinity of transcobalamin for vitamin B12” does not make sense. In addition, the sentence: “also after adjustment for other causes of hypercobalaminemia, i.e., myeloid blood malignancies” does not make sense. This is also a malignant disease.
Response: Thank you for this remark, indeed it is not clear. We provided pathomechanisms related to hypercobalaminemia: “ Several mechanisms can lead to high B12 levels: (i) excessive intake or administration, (ii) release from reservoirs (such as from the liver in cases of hepatic insufficiency and/or metastases), (iii) decreased vitamin B12 clearance, in cases of renal or hepatic insufficiency, (iv) an increase in vitamin B12 transporting proteins (transcobalamin or haptocorrin) produced by cancer cells, and (v) immune complexes, where vitamin B12 bound to immunoglobulins can falsely elevate measured levels [17,88].
The latter sentence was rewritten: Another study revealed that high cobalamin levels (>1000 ng/l) are associated with an increased risk of cancer, also after adjustment for other causes of hypercobalaminemia, i.e., liver diseases, chronic kidney failure, autoimmune or inflammatory diseases, and excessive B12 supplementation.
- Section 9 is potentially important but would need a more in depth and critical analysis of existing data.
Response: Thank you for the valuable comment, indeed the topic of CINP and its treatment is very interesting, regarding our paper, In section 9, we put all existing data on vitamin B supplementation in CIPN, which is scarce, but we rewrote this chapter to underline the studies results relevant for the clinical practice. Additionally, the paragraph is summarized with the current guidelines statement regarding CINP: “Overall, there is limited clinical evidence on using vitamin B12 in patients with CINP as a treatment or for prophylaxis, particularly in the absence of vitamin B12 deficiency. According to the current European Society of Medical Oncology (ESMO) guidelines for CINP, treatment with B vitamins cannot be recommended in this setting [104]. This topic requires further clinical research.”
- Section 10 is potentially important but would need a more in depth and critical analysis of existing data.
Thank you for highlighting the potential relevance of Section 10. We fully agree that the relationship between vitamin B12 and immunotherapy outcomes in cancer warrants deeper and more critical discussion, but unfortunately data on this topis is scarce. In response, we have revised Section 10 and added a comment: “Both deficiency and excess may be detrimental, albeit via different mechanisms. However, the precise immunological roles of B12 in cancer patients remain insufficiently characterized, and further research—especially well-controlled clinical trials—is necessary to determine whether active correction or modulation of B12 levels could improve immunotherapy outcomes.”
Reviewer 2 Report
Comments and Suggestions for Authors
- Intro
Opening phrase is a regurgitation/self-plagiarism of the abstract. Maybe not a problem, but it reads a little funny. "Occidental countries" is an interesting choice of words.
The section after citation [4,5] could do with some references for readers that want to learn about what is known about e.g. neuroprotective effects of B12 - add references here please. - Sources and uptake
Add the word "essential" to describe the vitamin.
Here is another example of self-plagiarism from the introduction. "at the cellular level..."
It is mentioned that cobalamin deficiency can lead to hematologic and neurologic symptoms, but it unclear how or why - please mention why these enzyme are specifically important to this group of symptoms.
This section ends with "as it may take up to 10-20 years". What does "this" refer to? Depletion of hepatic B12 pool? - Maybe add a table for the serum concentrations of B12 per age group?
"Since evaluating Vitamin B12 carries a risk of false results" - what type? False positive/negative? Is this specific to B12 or just meant in the sense that any assay carries a risk of false results? Is the assay that unreliable?
I guess the authors describe this in section 3.4, but I do not see a big relevance to B12 biology specifically. - Causes
This section can be restructured to emphasize that there are no specific causes of B12 deficiency that uniquely targets this vitamin but it is rather a general state of wasting and malnutrition and physiologic incapacity that also affects whole body nutritional state
Table 1 - please format so text is not centered and make an actual table instead of a list of words. It is too difficult to see the references and just looks unprofessional.
Table 2 - actually looks better, but for the love of science, please adjust the text to the left. - -
- Is there anything specific about tumors that would cause B12 deficiency? Or is it just the general poor clinical state, like cachexia and similar?
- -
- Remove the phrasing "a French study showed that" and similar. It is irrelevant in this context.
In general, it seems like hypercobalaminemia is most commonly associated with liver problems, and not that much with excessive intake - so please clarify whether hypercobalaminemia is most often asymptomatic or if it should be avoided because of its association with increased cancer incidence. - Results presented in table 3 should just emphasize that there is no clinical improvement after B12 therapy compared to placebo.
- -
- This is confusing:
Conclusion: "Vitamin B12 is a common finding in cancer patients" but in section 8 the others mention that hypercobalaminemia "is often observed in cancer patients".
The authors need to address this dual message throughout the review.
Author Response
- Intro
Opening phrase is a regurgitation/self-plagiarism of the abstract. Maybe not a problem, but it reads a little funny. "Occidental countries" is an interesting choice of words.
The section after citation [4,5] could do with some references for readers that want to learn about what is known about e.g. neuroprotective effects of B12 - add references here please.
Response: We thank the reviewer for this helpful observation. We have revised the opening sentence of the introduction to avoid repetition with the abstract and improve the flow. The phrase "Occidental countries" has been replaced with "USA" to ensure clarity.
Regarding the section following citations [4,5], we agree that additional references are warranted. In the introduction, we made a brief point about CIPN, while more details are described in the chapter about CIPN and vitamin B12.
- Sources and uptake
Add the word "essential" to describe the vitamin.
Here is another example of self-plagiarism from the introduction. "at the cellular level..."
Response: Thank you for the comment, the word essential was added and the expression at the cellular level was erased
- It is mentioned that cobalamin deficiency can lead to hematologic and neurologic symptoms, but it unclear how or why - please mention why these enzyme are specifically important to this group of symptoms.
Response: Thanks for this remark since we did not provide the cause-effect chain. Here are the explanations why vitamin B12 deficiency gives such symptoms: “Impaired methionine synthase inhibition results in demyelination of peripheral and central neurons, leading to the characteristic neurological manifestations of vitamin B12 deficiency [2,8]. The haematological effect of B12 deficiency is megaloblastic anaemia, which results from disruption of DNA synthesis (– see Chapter 5. Clinical Presentation of Vitamin B12 Deficiency).”
This section ends with "as it may take up to 10-20 years". What does "this" refer to? Depletion of hepatic B12 pool?
Response: This sentence was clarified in the text.
- Maybe add a table for the serum concentrations of B12 per age group?
Response: This could be an interesting point, but the reference is taken from a Danish study, and as authors we doubt that such a table could be used for worl-wide population, additionally every method and laboratory has different cut-offs. Therefore we did not prepare such a table.
"Since evaluating Vitamin B12 carries a risk of false results" - what type? False positive/negative? Is this specific to B12 or just meant in the sense that any assay carries a risk of false results? Is the assay that unreliable?
I guess the authors describe this in section 3.4, but I do not see a big relevance to B12 biology specifically.
Response: thank you for this comment, indeed the statement is misleading. We meant false negative results. But to be clear we erased this sentence from paragraph 3.3 and described the vitamin B12 assessment in the section 3.1 Serum vitamin B12 cut-off values and assessment limitations: “ The serum vitamin B12 concentration is the most common marker for assessing vitamin B12 deficiency due to its high availability and relatively low cost, but it carries some limitations. As already stated, low vitamin B12 serum concentration (ie <148 pmol/L) is the biochemical confirmation of vitamin B12 deficiency; concentration of serum vitamin B12 within normal range or even elevated does not exclude the deficiency [16,17]. This can be due to the patient’s marginal vitamin B12 status, which can already give symptoms, or analytical limitations, such as laboratory errors or altered B12-binding proteins. Total serum B12 is the sum of vitamin B12 bound to transporting proteins: transcobalamin and haptocorrin; therefore, an increased concentration of these proteins (common in cancer patients) leads to high levels of vitamin B12 [18].
- Causes
This section can be restructured to emphasize that there are no specific causes of B12 deficiency that uniquely targets this vitamin but it is rather a general state of wasting and malnutrition and physiologic incapacity that also affects whole body nutritional state
Response: We have modified this paragraph, and underlined that the causes of vitamin B12 deficiency in cancer patients can vary: “Vitamin B12 deficiency among cancer patients can be multifactorial. The deficiency can occur due to: (i) cancer site, especially if it involves organs essential for vitamin B12 absorption, such as the stomach, pancreas, or terminal ileum; (ii) undergoing gastrointestinal surgery; (iii) age above 65 years and associated comorbidities [27]. Importantly, malnutrition, advanced malignancy, and low performance status can lead to vitamin B12 deficiency or hypercobalaminemia [27]. Nonetheless, given its clinical relevance, it should not be overlooked and deserves appropriate evaluation. The possible causes of vitamin B12 deficiency are listed in Table 1, with a more detailed description of the deficiency causes relevant to cancer patients described below.”
Table 1 - please format so text is not centered and make an actual table instead of a list of words. It is too difficult to see the references and just looks unprofessional.
Table 2 - actually looks better, but for the love of science, please adjust the text to the left.
Response: You are right, the centered formatting looks bad. The text in the tables was adjusted to the left
- Is there anything specific about tumors that would cause B12 deficiency? Or is it just the general poor clinical state, like cachexia and similar?
Response: Thank you for this comment. We tried to make it clearer in this chapter what the reasons are for the different prevalence of vitamin B12 deficiency in different cancer subtypes. We made an introduction of the possible causes, as well as added more data on specific cancer subtypes and other factors affecting the prevalence :
"The prevalence of vitamin B12 deficiency among cancer patients depends on several factors. Aside from the cancer site (whether organs involved in vitamin B12 absorption are affected or not) and the treatments associated with cancer, which may differ between cancer types, the definition of vitamin B12 deficiency employed in the study (whether using a classical cut-off value or functional markers) impacts the prevalence rates. Additionally, the geographical regions described, along with their underlying vitamin B12 deficiency levels in the general population, translate into the percentage of patients entering oncological treatment with deficiency. Most studies cited below used the classical definition of vitamin B12 deficiency (serum vitamin B12 below the normal range, unless otherwise stated). "
- Remove the phrasing "a French study showed that" and similar. It is irrelevant in this context.
In general, it seems like hypercobalaminemia is most commonly associated with liver problems, and not that much with excessive intake - so please clarify whether hypercobalaminemia is most often asymptomatic or if it should be avoided because of its association with increased cancer incidence.
Response: Thank you for this comment, we removed the redundant phrasings. Additionally, we have underlined in the text that hypercobalaminemia is most often clinically asymptomatic, but this finding can be associated with increased cancer incidence and always needs further evaluation.
- Results presented in table 3 should just emphasize that there is no clinical improvement after B12 therapy compared to placebo.
Thank you for this comment, we modified Table 3, so the presented results are clearer for the reader.
Table 3. Studies on vitamin B12 supplementation in patients with chemotherapy-induced peripheral neuropathy (CIPN).
|
Author, year |
Type of Study |
Intervention |
Chemotherapy |
Result |
|
Schloss JM, Colosimo M, et al., 2015 [100] |
Case study |
vitamin B12 intramuscular |
Paclitaxel for breast cancer |
Vitamin B12 i.m and vitamin B complex improved CINP associated with B12 deficiency and paclitaxel chemotherapy in a patient with breast cancer |
|
Solomon L., 2016 [57] |
Retrospective study |
Vitamin B12 therapy (5 patients oral supplementation, 1 patient – intramuscular) |
No data |
From 241 cancer subjects, 3 patients with both functional vitamin B12 deficiency and neurologic abnormalities had clinical improvement after B12 therapy |
|
Schloss JM, Colosimo M, et al.,2017 [103] |
Randomized Clinical Trial |
Vitamin B group oral administration versus placebo |
Taxanes, oxaliplatin or vincristine for various neoplasms |
Patients with an oral B group vitamin supplementation did not have superior results to placebo group for the prevention of CIPN measured with total neuropathy score. Patients taking the B group vitamin perceived a reduction in sensory peripheral neuropathy in the Patient Neurotoxicity Questionnaire |
This is confusing:
Conclusion: "Vitamin B12 is a common finding in cancer patients" but in section 8 the others mention that hypercobalaminemia "is often observed in cancer patients".
The authors need to address this dual message throughout the review.
Response: Thank you for this remark, we don’t think that vitamin B12 deficiency and excess is mutually exclusive, but we tried to make it more clear throughout the review and in the conclusion section: “Abnormal vitamin B12 status—both deficiency and excess—is common among cancer patients. Reported deficiency rates range from 6% to 48%, depending on population and age, with the highest prevalence observed in patients with digestive tract cancers and elderly groups. Hypercobalaminemia occurs in approximately 17% of cancer patients and, in the general population, may serve as a marker of malignancy. [...]”